# Association between migration and severe maternal outcomes in high-income countries: Systematic review and meta-analysis

Maxime Eslier[1]*, Elie Azria[1,2], Konstantinos Chatzistergiou[2], Zelda Stewart[1], Agnès Dechartres[3], Catherine Deneux-Tharaux[1]

1 Université Paris Cité, CRESS, Obstetrical Perinatal and Pediatric Epidemiology Research Team, EPOPé, INSERM, INRAE, Paris, France, 2 Maternity Unit, Paris Saint Joseph Hospital, FHU PREMA, Paris, France, 3 Sorbonne Université, INSERM, Institut Pierre Louis d'Epidémiologie et de Santé Publique, AP-HP. Sorbonne Université, Hôpital Pitié-Salpêtrière, Département de Santé Publique, Paris, France

☯ These authors contributed equally to this work.
* maxime.eslier@inserm.fr

**Data Availability Statement:** All relevant data are within the manuscript and its Supporting Information files.

**Funding:** The authors received no specific funding for this work.

## Abstract

### Background

Literature focusing on migration and maternal health inequalities is inconclusive, possibly because of the heterogeneous definitions and settings studied. We aimed to synthesize the literature comparing the risks of severe maternal outcomes in high-income countries between migrant and native-born women, overall and by host country and region of birth.

### Methods and findings

Systematic literature review and meta-analysis using the Medline/PubMed, Embase, and Cochrane Library databases for the period from January 1, 1990 to April 18, 2023. We included observational studies comparing the risk of maternal mortality or all-cause or cause-specific severe maternal morbidity in high-income countries between migrant women, defined by birth outside the host country, and native-born women; used the New-castle–Ottawa scale tool to assess risk of bias; and performed random-effects meta-analyses. Subgroup analyses were planned by host country and region of birth.

The initial 2,290 unique references produced 35 studies published as 39 reports covering Europe, Australia, the United States of America, and Canada. In Europe, migrant women had a higher risk of maternal mortality than native-born women (pooled risk ratio [RR], 1.34; 95% confidence interval [CI], 1.14, 1.58; $p < 0.001$), but not in the USA or Australia. Some subgroups of migrant women, including those born in sub-Saharan Africa (pooled RR, 2.91; 95% CI, 2.03, 4.15; $p < 0.001$), Latin America and the Caribbean (pooled RR, 2.77; 95% CI, 1.43, 5.35; $p = 0.002$), and Asia (pooled RR, 1.57, 95% CI, 1.09, 2.26; $p = 0.01$) were at higher risk of maternal mortality than native-born women, but not those born in Europe or in the Middle East and North Africa. Although they were studied less often and with heterogeneous definitions of outcomes, patterns for all-cause severe maternal morbidity and maternal intensive care unit admission were similar. We were unable to take into account other

**Competing interests:** I have read the journal's policy and EA have the following competing interests: EA declared participation in the Data safety Monitoring board of the Betadose trial, Member of the advisory board of the Mamaact Trial and Head of the commission "Health inequalities" of the French College of Obstetrician and Gynecologists (CNGOF).

**Abbreviations:** CI, confidence interval; ICD, International Classification of Diseases; ICU, intensive care unit; RR, risk ratio; ROAM, Reproductive Outcomes and Migration; USA, United States of America; WHO, World Health Organization.

social factors that might interact with migrant status to determine maternal health because many of these data were unavailable.

## Conclusions

In this systematic review of the existing literature applying a single definition of "migrant" women, we found that the differential risk of severe maternal outcomes in migrant versus native-born women in high-income countries varied by host country and region of origin. These data highlight the need to further explore the mechanisms underlying these inequities.

## Trial Registration

PROSPERO CRD42021224193.

### Why was this study done?

- Some studies conducted in high-income countries report that the risk of maternal mortality and severe maternal morbidity is higher for migrant than native-born women, while other studies do not.
- Whether this heterogeneity is related to differences in the definition and measurement of migration and of maternal outcomes or to real differences between settings remains unclear.

### What did the researchers do and find?

- In this literature review including 35 studies, we found that, in high-income countries, the differential risk of severe maternal outcomes in migrant women, defined as born outside the host country, compared to native-born women, varied by the host country and the migrant women's region of birth.
- In Europe, migrant women were generally at higher risk of severe maternal outcomes than native-born women, whereas the risks for migrant women did not differ significantly from those for native-born women in United States of America or Australia.
- Among migrant women, those born in sub-Saharan Africa, in Latin America and the Caribbean, or in Asia were at higher risk of severe maternal outcomes than their native-born counterparts, while those born in Europe or in the Middle East and North Africa were not.

### What do these findings mean?

- When a single definition of "migrant" women was applied, the differential risk of severe maternal outcomes in migrant versus native-born women in high-income countries varied by host country and region of origin.
- These data highlight the need to further explore the mechanisms underlying these inequities.
- Future studies should use harmonized definitions for all-cause and cause-specific severe maternal morbidity and take into account other social factors such as race/

ethnicity, migrant women's administrative status, and economic factors that may interact with migration to understand the inequalities in maternal health between migrant and native-born women.

## Introduction

Economic crises, wars, natural disasters, and increased inequalities between countries have generated significant migration waves. Over the past decade, the migrant population has increased by 23% in countries belonging to the Organisation for Economic Co-operation and Development [1] and by 28% in the European Union [2]. These movements, involving large proportions of young adults, women in particular [2], result in increasing the proportion of births to foreign-born women in their host countries. In Europe, every fourth birth is to a foreign-born mother, although this rate varies substantially across countries [3,4]. This trend presents organizational and cost challenges for the host health systems, especially as migrant women appear to be more vulnerable and to have greater social and health needs than native-born women [5,6].

Because the scientific literature on the association between migration and severe maternal outcomes is heterogeneous and inconclusive, it cannot accurately inform public policies or healthcare provision [7]. Some studies in high-income countries have reported higher risks of maternal mortality and severe maternal morbidity among migrant women than native-born women [8–13], while others have not [14–17]. Whether this heterogeneity is related to differences in definitions and methods of measuring migration and maternal outcomes or to real differences between settings remains unclear. Studies have defined migrant women according to their race or ethnicity [18–23], nationality [24–27], or their birthplace [14,16,17,28–33]. A few others have used other criteria, such as the Human Development Index of the country of origin [34], length of residency in the host country, or legal status [35]. The Reproductive Outcomes and Migration (ROAM) collaboration and the EURO-PERISTAT project recommended that maternal country of birth be used to study immigrants' perinatal health [36]. The severe maternal outcomes targeted and their definitions also vary greatly across studies. Moreover, because much of the available literature is based on routinely collected data coded by the International Classification of Diseases (ICD), severe maternal outcomes might well be misclassified [14,15,17,37–43]. A previous review and meta-analysis of data from Western European countries between 1970 and 2013 showed that migrant women have a risk of maternal mortality twice that of women born in the host country [8]. Its results were nonetheless limited by the important heterogeneity in the definitions of migrant women across studies and the inclusion of old data. Another systematic review also conducted on European studies compared the risk of nonsevere maternal outcomes between 2007 and 2017 for specific subgroups of migrant women only—asylum seekers and undocumented migrant women in Europe. It reported that the risk of maternal mortality and all-cause severe maternal morbidity was higher among asylum seekers than native-born women [35].

Applying a single definition of migrant women, in accordance with international recommendations, we aimed to conduct a systematic review and meta-analysis of available information about the risk of severe maternal outcomes among migrant and native-born women in high-income countries. Differential risk was compared overall and by both host country and migrant women's region of birth.

## Methods

### Search strategy and selection criteria

This systematic review and meta-analysis examines studies of the association between migration and severe maternal outcomes in high-income countries. The review protocol was registered with the PROSPERO International Prospective Register of Systematic Reviews, on its website (CRD42021224193), and this article is reported according to the Preferred Reporting Items for Systematic Review and Meta-analysis (PRISMA) guidelines [44].

We searched Medline via PubMed, Embase, and Cochrane Library without language restriction. The search algorithm included relevant Medical Subject Headings (MeSH)/Embase Medical Headings (EMTREE) and free text words combined by Boolean operators for migrant and selected severe maternal outcomes (list below) (S1–S3 Tables). We also screened the reference lists of previous systematic reviews and meta-analyses and of all included studies for any additional references.

We included observational studies published between January 1, 1990 and April 18, 2023, and comparing the risk of severe maternal outcomes between migrant women and native-born women in high-income countries, according to the World Bank classification. Systematic reviews and meta-analyses were included in the first selection by titles and abstracts to allow backward snowballing but excluded in the second selection. Case–control and case studies were excluded. The exclusion of studies before 1990 aimed to limit the heterogeneity in migrant women's characteristics and host countries' integration policies that could be caused by too long a study period.

Our review included studies that defined migrant women as women born outside the host country and compared them with women born in the host country, in accordance with international recommendations [36,45,46]. The severe maternal outcomes studied were maternal mortality, all-cause severe maternal morbidity, and cause-specific severe maternal morbidity, during pregnancy and up to 1 year after delivery. The specific causes of severe maternal morbidity included were severe postpartum hemorrhage, eclampsia, severe sepsis, and uterine rupture, as well as maternal intensive care unit (ICU) admission and near misses, as defined by the World Health Organization (WHO) [47]. This review did not consider maternal mental health outcomes because we believe they require a specific assessment and because a systematic review on this topic was registered as underway when we started (PROSPERO, CRD42021226291) and finally published [48].

We used Covidence systematic review software (Veritas Health Innovation, Melbourne, Victoria, Australia) to screen and extract data. Two reviewers (ME and KC) independently screened first the titles and abstracts of the retrieved studies to exclude those that were irrelevant and then the full text of the remaining studies to assess eligibility for inclusion. Disagreements were resolved through discussion with the help, if necessary, of a third author (EA or CDT) to reach a consensus. Reasons for exclusion were recorded. The corresponding author of unavailable articles was contacted by email to request the full text version.

### Data analysis

The same reviewers (ME and KC) independently extracted the following characteristics for each included study: general study information (authors' names, year of publication, journal, language, funding); study characteristics (study design, population coverage, inclusion period, host country, and data source for the exposure of interest and for outcomes); characteristics of the study population (total number of women, inclusion/exclusion criteria); outcomes evaluated (mortality and/or all-cause severe maternal morbidity and/or cause-specific severe

maternal morbidity); number of cases overall, in each subgroup and for each outcome of interest; and items to assess methodological quality. We contacted corresponding authors to obtain additional data not available in the original publication.

Two authors (ME and KC) used the Newcastle–Ottawa scale for observational studies to assess the methodological quality of each study and based their assessment on the following categories of items: selection of study groups, comparability of groups, and ascertainment of the outcome of interest [49]. The threshold for defining a high-quality study was a score $\geq 7$ [50].

An unadjusted risk ratio for each outcome was estimated with the number of events and of migrant women and native-born women reported in each study. We did not estimate adjusted risk ratios because this study examined the differential risk between migrant women and native-born women and not causal mechanisms. Between-study heterogeneity was explored by visual examination of the forest plots and by the heterogeneity test and $I^2$ statistic with its 95% confidence interval (CI) [51,52]. Because heterogeneity was expected on scientific grounds, we used random-effects meta-analysis models with the DerSimonian and Laird method [51] and decided, when relevant, not to provide an overall assessment and to combine data only within subgroups. Those subgroups were defined according to the host country (European countries, the United States of America (USA), Canada, and Australia), the migrant women's regions of birth (Europe, the Middle East and North Africa, sub-Saharan Africa, Latin America and the Caribbean, and Asia and the Pacific), the migrant women's legal status (migrant women with host country nationality, legally resident migrant women without host country nationality, and undocumented migrant women), and the Newcastle-Ottawa scale score. We tested the interaction between the exposure of interest and the variables defining subgroups. The considerable heterogeneity in definitions of cause-specific severe maternal morbidity prevented the performance of any quantitative meta-analysis for these outcomes.

We assessed small-study effects with funnel plots when there were sufficient studies ($n > 10$).

Analyses were done with Review Manager Software (version 5.4.1).

## Results

### Characteristics and methodological quality assessment

Of the 2,290 records screened, 185 were relevant for full-text review and 39 were finally included (Fig 1). The appendix provides a list of excluded studies with the reasons for their exclusion (S4 Table). The leading reason for exclusion was the absence of a definition of migration based on region of birth, followed by outcomes outside the scope of our review.

Table 1 shows the characteristics of the 35 studies from Europe, Australia, the USA, and Canada that furnished 39 reports. Because 4 studies yielded several reports covering different outcomes or different subgroups of migrant women, we included 39 reports overall. The median year of publication was 2017 (range 2008 to 2022). Half the studies used the ICD or procedure codes to define their outcomes (Tables 1 and S5). The assessment of the quality of the studies included is presented in the appendix (S6 Table). According to the Newcastle–Ottawa scale score, only 1 study had a low methodological quality that limited the relevance of the subgroup analysis (Tables 1 and S6). Because only 3 studies contained information about legal status, we did not perform the corresponding subgroup analysis (Table 1).

### Maternal mortality

Seventeen highly heterogeneous studies included data on maternal mortality (S1 Fig). In European host countries, migrant women were at higher risk of maternal mortality than native-

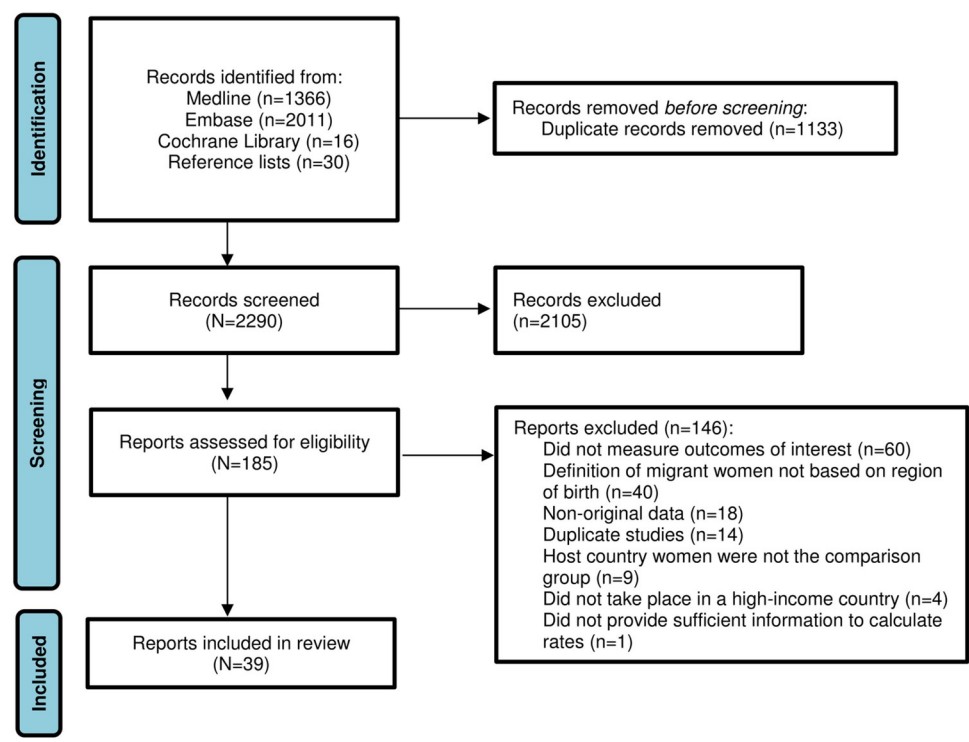

**Fig 1. Study selection.**

born women (pooled risk ratio [RR] 1.34; 95% CI, 1.14, 1.58; $p < 0.001$; $I^2 = 58\%$; 95% CI; 18, 79), with RRs ranging from 0.77 (95% CI, 0.15, 3.80) to 2.01 (95% CI, 1.53, 2.65), whereas no excess risk was observed in the USA or Australia (p for interaction $< 0.001$) (Fig 2). Some subgroups of migrant women were at higher risk of maternal mortality than their native-born counterparts. They included women born in sub-Saharan Africa (pooled RR, 2.91; 95% CI, 2.03, 4.15; $p < 0.001$; $I^2 = 57\%$; 95% CI, 0, 83), in Latin America and the Caribbean (pooled RR, 2.77; 95% CI, 1.43, 5.35; $p = 0.002$; $I^2 = 82\%$; 95% CI; 53, 93), and in Asia (pooled RR, 1.57; 95% CI, 1.09, 2.26; $p = 0.01$; $I^2 = 31\%$; 95% CI, 0, 74). Women born in Europe or in the Middle East and North Africa, on the other hand, were not at higher risk (p for interaction $< 0.001$) (Fig 3). Among European countries, the United Kingdom, Denmark, Norway, and Spain presented a particular pattern, with no significant excess risk of maternal mortality for migrant women considered globally (Fig 2), but specific subgroups studied according to their region of birth did show such an excess risk compared to native-born women (Fig 3).

## All-cause severe maternal morbidity

Similar patterns were found for both all-cause severe maternal morbidity and maternal ICU admission. The meta-analysis of the former included 10 studies comparing migrant women and native-born women with substantial heterogeneity (Figs 4, 5 and S2). The pooled relative risk of all-cause severe maternal morbidity in migrant women compared to native-born women was 1.19 (95% CI, 0.95, 1.49; $p = 0.1$; $I^2 = 87\%$; 95% CI, 69, 95) in European host countries and 0.99 (95% CI, 0.95, 1.03; $p = 0.7$; $I^2 = 83\%$; 95% CI, 64, 92) in the USA and Australia (p for interaction $< 0.001$) (Fig 4). Among migrant women, those born in sub-Saharan Africa (pooled RR, 1.54; 95% CI, 1.26, 1.88; $p < 0.001$; $I^2 = 88\%$; 95% CI, 72, 95) or in Latin America and the Caribbean (pooled RR, 1.14; 95% CI, 1.05, 1.25; $p = 0.003$; $I^2 = 51\%$; 95% CI, 0, 84)

**Table 1. Characteristics of included studies.**

| Total = 35 studies | |
|---|---|
| First year of inclusion period, median (min-max) | 2007 (1988–2016) |
| Last year of inclusion period, median (min-max) | 2012 (2005–2019) |
| **Host country, n (%)** | |
| Australia | 7 (20) |
| United States of America | 5 (14) |
| France | 4 (11) |
| Canada | 4 (11) |
| the Netherlands | 3 (9) |
| United Kingdom | 3 (9) |
| Sweden | 2 (6) |
| Denmark | 2 (6) |
| Germany | 2 (6) |
| Norway | 1 (3) |
| Spain | 1 (3) |
| Italy | 1 (3) |
| Multicountry | 1 (3) |
| **Study population, n (%)** | |
| National | 21 (60) |
| Regional | 9 (26) |
| Mixed (National and Regional) | 1 (3) |
| Multicenter | 2 (6) |
| Single-center | 3 (9) |
| **Study design, n (%)** | |
| Cohort study | 19 (54) |
| Cross-sectional study | 17 (49) |
| **Source of information regarding exposure of interest\*, n (%)** | |
| Birth and/or death certificates | 29 (83) |
| Birth register | 4 (11) |
| Interview | 1 (3) |
| Medical file | 1 (3) |
| Self-administered questionnaire | 1 (3) |
| **Legal status available, n (%)** | 3 (9) |
| **Reported outcome, n (%)** | |
| Maternal mortality | 17 (49) |
| All-cause severe maternal morbidity | 10 (29) |
| Maternal near miss | 3 (9) |
| Maternal intensive care unit admission | 5 (14) |
| Eclampsia | 4 (11) |
| Severe postpartum hemorrhage | 6 (17) |
| Severe sepsis | 4 (11) |
| Uterine rupture | 8 (23) |
| **Definition of outcome, n (%)** | |
| Clinical criteria specifically collected | 18 (51) |
| Existing ICD or procedure codes | 18 (51) |
| **Source of information regarding outcome, n (%)** | |
| Medico-administrative data | 15 (43) |
| Medical file | 19 (54) |

(*Continued*)

**Table 1.** (Continued)

| Total = 35 studies | | |
|---|---|---|
| | Birth register | 2 (6) |
| **Newcastle–Ottawa scale score**, median (min-max) | | 7 (6–8) |

*Migrant women defined by region of birth.

were at higher risk of all-cause severe maternal morbidity than their native-born counterparts, while those born in Europe or the Middle East and North Africa were not (p for interaction < 0.001) (Fig 5).

## Cause-specific severe maternal morbidity

Only a few studies considering specific causes or conditions of severe maternal morbidity were available (Figs 6–11). The synthesis of their results was difficult given the persisting heterogeneity in outcome definitions, except for maternal ICU admission (Fig 6). Three of 5 studies reported that migrant women were at higher risk of maternal ICU than native-born women with relative risks ranging from 1.34 (95% CI, 1.23, 1.46) to 1.97 (95% CI, 1.55, 2.52) (Figs 6 and S3). Migrant women born in sub-Saharan Africa, in Latin America and the Caribbean, and in Asia were also at higher risk of maternal ICU admission than native-born women, but those born in Europe or in the Middle East and North Africa were not (p for interaction < 0.001) (S4 Fig).

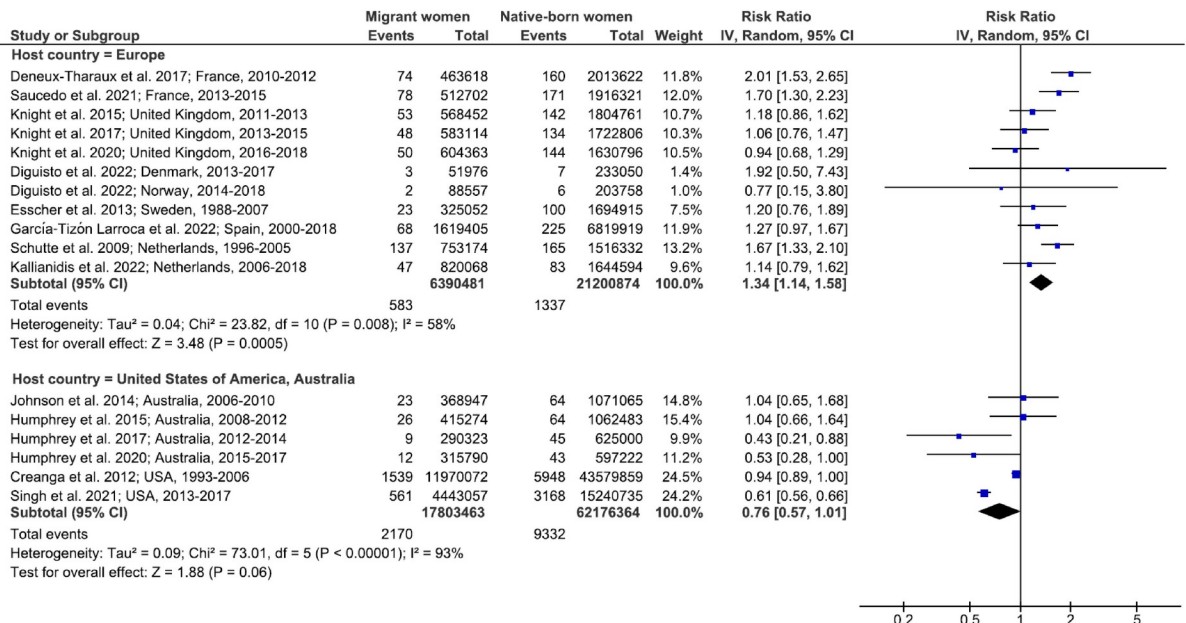

**Fig 2. Maternal mortality in migrant women and native-born women, stratified by host country.** Unadjusted RRs are random-effects estimates calculated by the DerSimonian and Laird method. The data markers show the unadjusted RRs with their 95% CIs. The size of the data markers indicates the weight of the study. Diamonds show the pooled unadjusted RRs. The CI is shown with lines. References: Deneux-Tharaux et al. 2017 [29]; Saucedo et al. 2021 [53]; Knight et al. 2015 [54]; Knight et al. 2017 [55]; Knight et al. 2020 [56]; Diguisto et al. 2022 [57]; Esscher et al. 2013 [28]; Garcia-Tizon Larroca et al. 2022 [58]; Schutte et al. 2009 [32]; Kallianidis et al. 2022 [59]; Johnson et al. 2014 [60]; Humphrey et al. 2015 [61]; Humphrey et al. 2017 [62]; Humphrey et al. 2020 [63]; Creanga et al. 2012 [16]; Singh et al. 2021 [14]. CI, confidence interval; df, degrees of freedom; IV, inverse variance; RR, risk ratio.

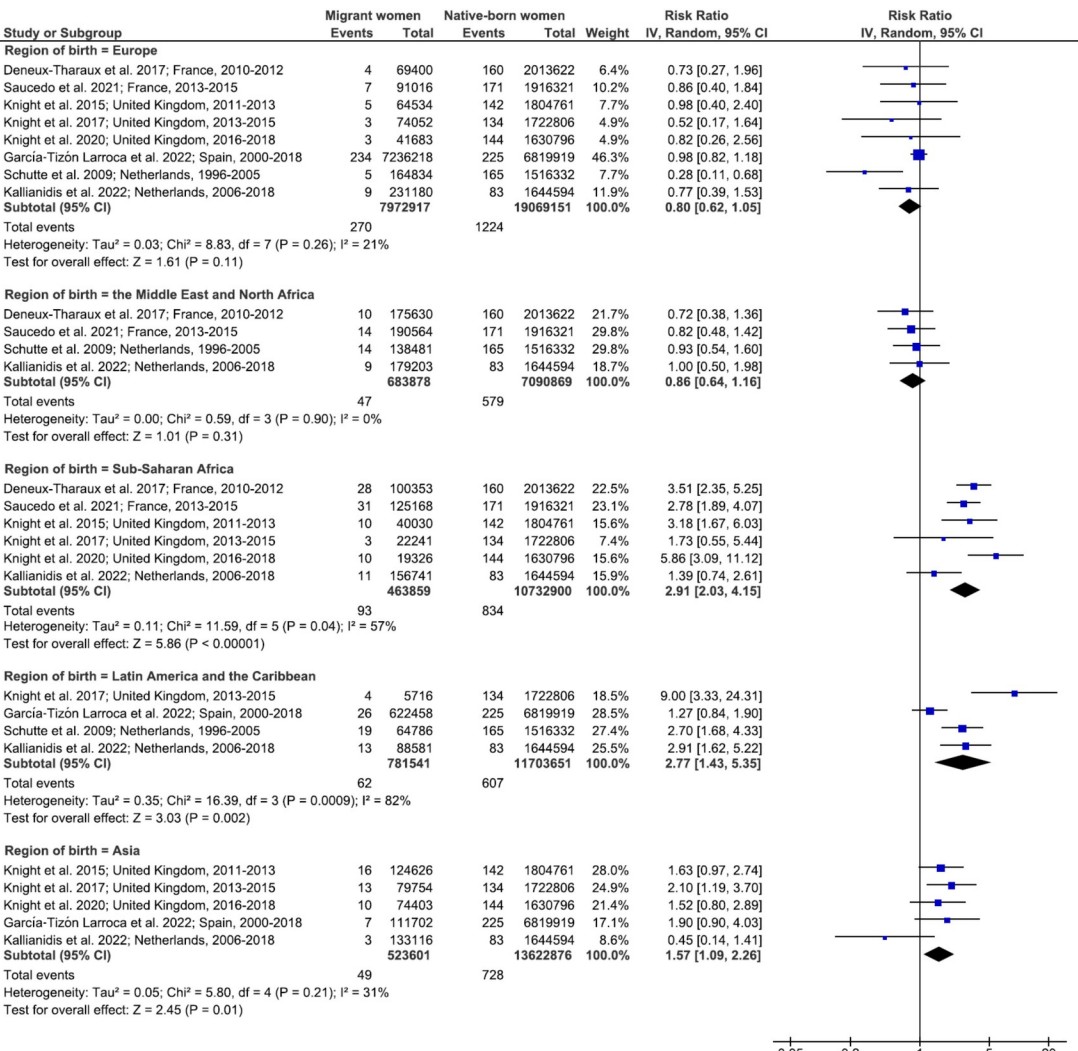

**Fig 3. Maternal mortality in migrant women and native-born women, stratified by migrant women's region of birth.**
Unadjusted RRs are random-effects estimates calculated by the DerSimonian and Laird method. The data markers show the unadjusted RRs with their 95% CIs. The size of the data markers indicates the weight of the study. Diamonds show the pooled unadjusted RRs. The CI is shown with lines. References: Deneux-Tharaux et al. 2017 [29]; Saucedo et al. 2021 [53]; Knight et al. 2015 [54]; Knight et al. 2017 [55]; Knight et al. 2020 [56]; Garcia-Tizon Larroca et al. 2022 [58]; Schutte et al. 2009 [32]; Kallianidis et al. 2022 [59]. CI, confidence interval; df, degrees of freedom; IV, inverse variance; RR, risk ratio.

A funnel plot was performed only for maternal mortality, as it was the only outcome for which a sufficient number of studies was available. It suggests a small-study effect (S5 Fig), which may be partly explained by the heterogeneity across host countries (S6 Fig).

## Discussion

In high-income countries, the differential risk of severe maternal outcomes between migrant and native-born women, defined as women born outside and in the host country, varied by both host country and the migrant women's region of birth. Migrant women in European host countries were at higher risk of severe maternal outcomes than native-born women, while migrant women in the USA and Australia did not differ from those born there for such outcomes. Migrant women born in sub-Saharan Africa, in Latin America and the Caribbean, or

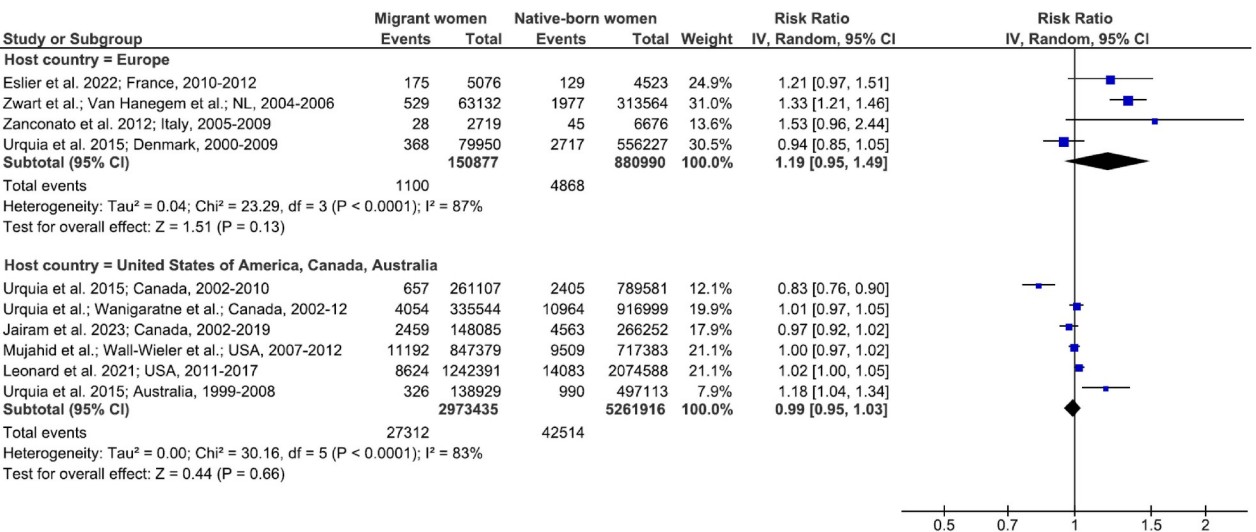

**Fig 4. All-cause severe maternal morbidity in migrant women and native-born women, stratified by host country.** Unadjusted RRs are random-effects estimates calculated by the DerSimonian and Laird method. The data markers show the unadjusted RRs with their 95% CIs. The size of the data markers indicates the weight of the study. Diamonds show the pooled unadjusted RRs. The CI is shown with lines. References: Eslier et al. 2022 [64]; Zwart et al. 2008 [9], 2010 [65], and 2011 [66]; Van Hanegem et al. 2011 [67]; Zanconato et al. 2012 [33]; Urquia et al. 2015 [40] and 2017 [15]; Wanigaratne et al. 2015 [41]; Jairam et al. 2023 [68]; Mujahid et al. 2021 [17]; Wall-Wieler et al. 2020 [42]; Leonard et al. 2021 [37]. CI, confidence interval; df, degrees of freedom; IV, inverse variance; RR, risk ratio.

in Asia were at higher risk of severe maternal outcomes than the women born in their host countries, but those born in Europe or in the Middle East and North Africa were not.

Our systematic review shows that migrant women living in European host countries had a higher risk of severe maternal outcomes than the native-born women, while migrant women in the USA and Australia did not. The social structure of the native-born groups is a hypothesis that might explain this disparity between these high-income settings. In the USA and Australia, the groups with the highest maternal mortality and morbidity rates are, respectively, Black [75] and indigenous [60–63] women, both included in the native-born reference group. The maternal mortality ratio is from 3 to 6 times higher in Black than in White women in the USA [76] and in indigenous versus nonindigenous women in Australia [60–63]. Furthermore, the overall maternal mortality ratio in the USA is more than twice as high as in Europe, and this ratio for native-born US White women is double that for native-born Europeans [53,55,76]. In both the USA and Australia, therefore, the subgroups at risk of severe maternal complications appear to be related more to race or ethnicity than to migrant women's region of birth. Thus, in these contexts, the comparison between "native" and "foreign-born" populations does not take into account the colonial history and migratory backgrounds of the now "native" populations that played a role in determining these groups' social position and their health status.

Another interesting finding is that migrant women born in sub-Saharan Africa, in Latin America and the Caribbean, and in Asia were at higher risk of severe maternal outcomes than the corresponding native-born women of the host country, although migrant women born in Europe or in the Middle East and North Africa were not. Notably, the particular pattern observed in the United Kingdom, with no significant excess risk in migrant women versus native-born women globally, when we did not consider region of birth, highlights the importance of studying migration according to this factor. One hypothesis for this differential risk among subgroups of migrant women is that the specific migration contexts and paths of these subgroups do not carry the same risks. Migrant women born in sub-Saharan Africa, in Latin

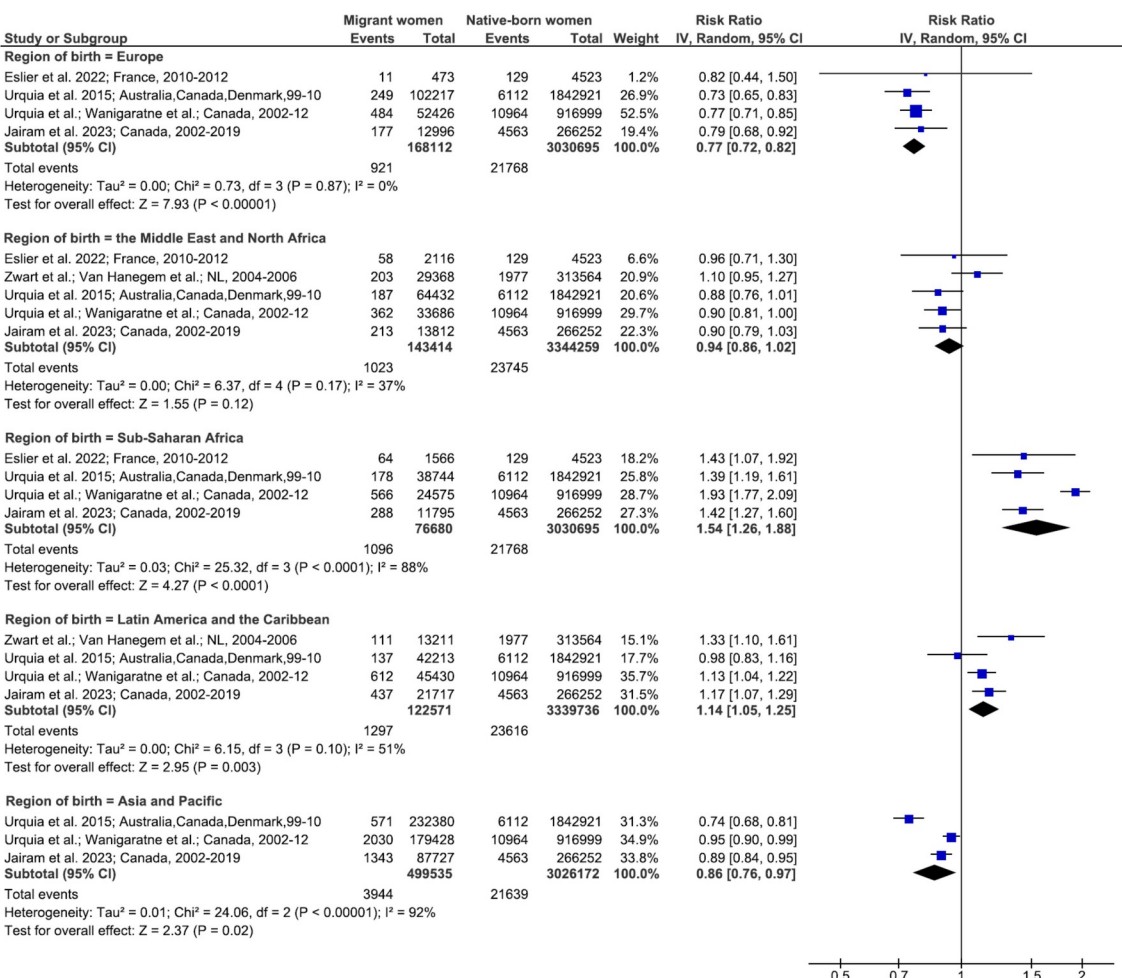

**Fig 5. All-cause severe maternal morbidity in migrant women and native-born women, stratified by migrant women's region of birth.** Unadjusted RRs are random-effects estimates calculated by the DerSimonian and Laird method. The data markers show the unadjusted RRs with their 95% CIs. The size of the data markers indicates the weight of the study. Diamonds show the pooled unadjusted RRs. The CI is shown with lines. References: Eslier et al. 2022 [64]; Urquia et al. 2015 [40] and 2017 [15]; Wanigaratne et al. 2015 [41]; Jairam et al. 2023 [68]; Zwart et al. 2008 [9], 2010 [65], and 2011 [66]; Van Hanegem et al. 2011 [67]. CI, confidence interval; df, degrees of freedom; IV, inverse variance; RR, risk ratio.

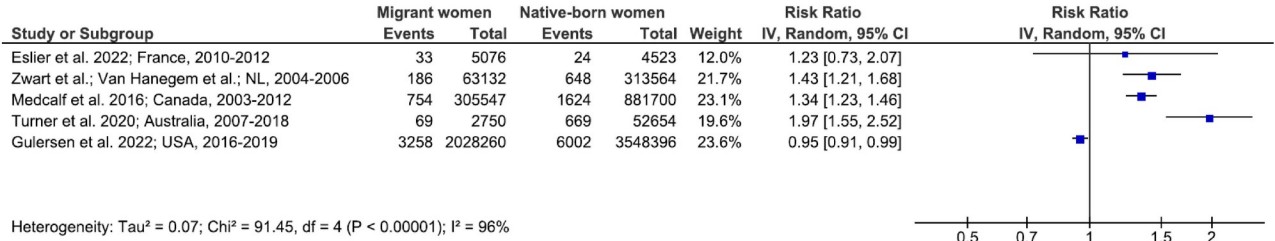

**Fig 6. Maternal ICU admission in migrant women and native-born women.** The squares show the unadjusted RRs with their 95% CIs. The size of the squares indicates the weight of the study. The CI is shown with lines. References: Eslier et al. 2022 [64]; Zwart et al. 2008 [9], 2010 [65] and 2011 [66]; Van Hanegem et al. 2011 [67]; Medcalf et al. 2016 [38]; Turner et al. 2020 [69]; Gulersen et al. 2022 [70]. CI, confidence interval; df, degrees of freedom; ICU, intensive care unit; IV, inverse variance; RR, risk ratio.

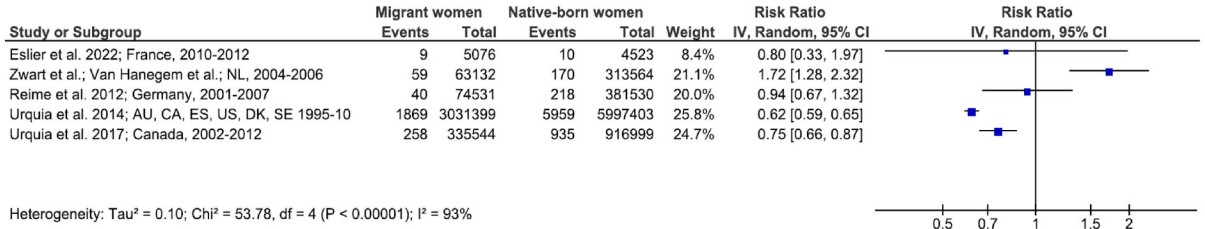

**Fig 7. Near misses in migrant women and native-born women.** The squares show the unadjusted RRs with their 95% CIs. The size of the squares indicates the weight of the study. The CI is shown with lines. References: David et al. 2019 [71]; Wahlberg et al. 2013 [72]; Reime et al. 2012 [73]. CI, confidence interval; df, degrees of freedom; IV, inverse variance; RR, risk ratio.

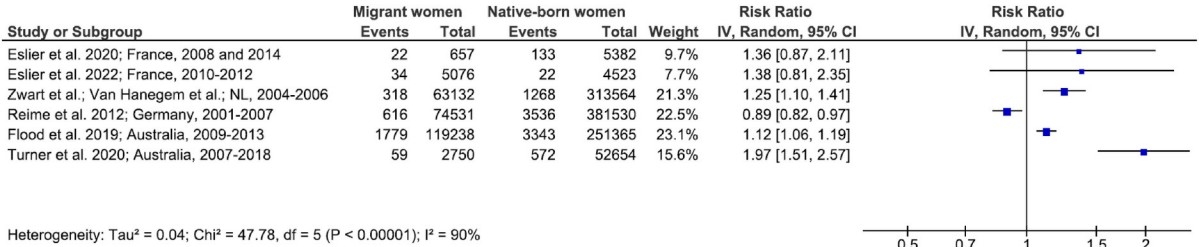

**Fig 8. Eclampsia in migrant women and native-born women.** The squares show the unadjusted RRs with their 95% CIs. The size of the squares indicates the weight of the study. The CI is shown with lines. References: Eslier et al. 2022 [64]; Zwart et al. 2008 [9], 2010 [65] and 2011 [66]; Van Hanegem et al. 2011 [67]; Reime et al. 2012 [73]; Urquia et al. 2014 [39] and 2017 [15]. CI, confidence interval; df, degrees of freedom; IV, inverse variance; RR, risk ratio.

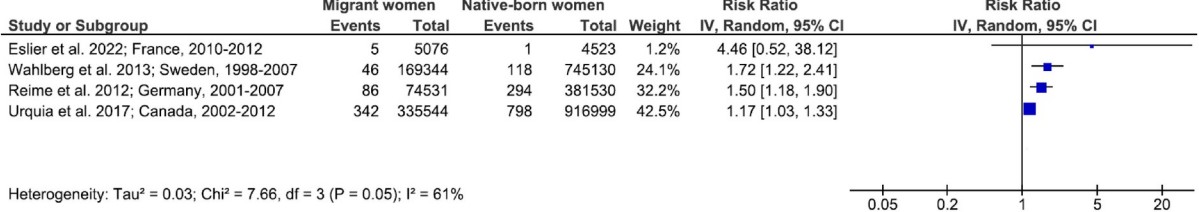

**Fig 9. Severe postpartum hemorrhage in migrant women and native-born women.** The squares show the unadjusted RRs with their 95% CIs. The size of the squares indicates the weight of the study. The CI is shown with lines. References: Eslier et al. 2020 [31]; Eslier et al. 2022 [64]; Zwart et al. 2008 [9], 2010 [65] and 2011 [66]; Van Hanegem et al. 2011 [67]; Reime et al. 2012 [73]; Flood et al. 2019 [74]; Turner et al. 2020 [69]. CI, confidence interval; df, degrees of freedom; IV, inverse variance; RR, risk ratio.

| Study or Subgroup | Migrant women Events | Total | Native-born women Events | Total | Weight | Risk Ratio IV, Random, 95% CI |
|---|---|---|---|---|---|---|
| Eslier et al. 2022; France, 2010-2012 | 5 | 5076 | 1 | 4523 | 1.2% | 4.46 [0.52, 38.12] |
| Wahlberg et al. 2013; Sweden, 1998-2007 | 46 | 169344 | 118 | 745130 | 24.1% | 1.72 [1.22, 2.41] |
| Reime et al. 2012; Germany, 2001-2007 | 86 | 74531 | 294 | 381530 | 32.2% | 1.50 [1.18, 1.90] |
| Urquia et al. 2017; Canada, 2002-2012 | 342 | 335544 | 798 | 916999 | 42.5% | 1.17 [1.03, 1.33] |

Heterogeneity: Tau² = 0.03; Chi² = 7.66, df = 3 (P = 0.05); I² = 61%

**Fig 10. Severe sepsis in migrant women and native-born women.** The squares show the unadjusted RRs with their 95% CIs. The size of the squares indicates the weight of the study. The CI is shown with lines. References: Eslier et al. 2022 [64]; Wahlberg et al. 2013 [72]; Reime et al. 2012 [73]; Urquia et al. 2017 [15]. CI, confidence interval; df, degrees of freedom; IV, inverse variance; RR, risk ratio.

| Study or Subgroup | Migrant women Events | Migrant women Total | Native-born women Events | Native-born women Total | Weight | Risk Ratio IV, Random, 95% CI |
|---|---|---|---|---|---|---|
| Eslier et al. 2020; France, 2008 and 2014 | 0 | 657 | 8 | 5382 | 0.8% | 0.48 [0.03, 8.33] |
| Eslier et al. 2022; France, 2010-2012 | 8 | 5076 | 2 | 4523 | 2.4% | 3.56 [0.76, 16.78] |
| Wahlberg et al. 2013; Sweden, 1998-2007 | 233 | 169344 | 847 | 745130 | 25.5% | 1.21 [1.05, 1.40] |
| Zwart et al.; Van Hanegem et al.; NL, 2004-2006 | 48 | 63132 | 170 | 313564 | 19.2% | 1.40 [1.02, 1.93] |
| Urquia et al. 2015; Australia,Canada,Denmark,99-10 | 1714 | 1842921 | 378 | 479986 | 26.5% | 1.18 [1.06, 1.32] |
| Urquia et al. 2017; Canada, 2002-2012 | 238 | 335544 | 862 | 916999 | 25.6% | 0.75 [0.65, 0.87] |

Heterogeneity: Tau² = 0.06; Chi² = 33.93, df = 5 (P < 0.00001); I² = 85%

**Fig 11. Uterine rupture in migrant women and native-born women.** The squares show the unadjusted RRs with their 95% CIs. The size of the squares indicates the weight of the study. The CI is shown with lines. References: Eslier et al. 2020 [31] and 2022 [64]; Wahlberg et al. 2013 [72]; Zwart et al. 2008 [9], 2010 [65] and 2011 [66]; Van Hanegem et al. 2011 [67]; Urquia et al. 2015 [40] and 2017 [15]. CI, confidence interval; df, degrees of freedom; IV, inverse variance; RR, risk ratio.

America and the Caribbean, and in Asia have usually lived in the host country for less time and are more frequently disadvantaged by a language barrier, lack of legal status, social isolation, and poor housing conditions, compared with other categories of migrant women [5,28]. These cofactors may result in a more difficult access to healthcare system, especially prenatal care, which is known to be inadequate more often, in both quantity and quality, among these geographical subgroups [77,78]. Researchers have explored the association between severe maternal outcomes and other categorizations of migrant women's native countries, besides geography. One study using the World Bank classification of economies found a higher risk of maternal near misses in migrant women from low-income countries and not in those from middle- or high-income countries, compared with women born in Sweden [72]. A Spanish study used the Human Development Index of countries to report that maternal death rates were higher for migrant women from countries with the lowest Human Development Index [58]. We could not include these analyses in our review since each was unique in its categorization. These approaches, however, provide interesting insights into the causal mechanisms of health disparities among migrant women.

Another explanatory hypothesis for the differential risk we found according to the migrant women's region of birth is the possibility of discrimination against some subgroups of migrant women with physical (e.g., black skin) or cultural singularities that might activate bias among healthcare professionals and lead to differential care [13,79–81]. These discriminations may originate from individual attitudes through explicit or implicit bias but could also be driven by structural racism [82,83]. An approach that focuses on the migration issue and takes the fact of being born abroad as the only exposure variable cannot by itself provide an insight into the mechanisms of social inequalities in health between migrant and native-born women. This approach is necessary and, in our opinion, justifies the studies thus far conducted in this way and included in this systematic review. It cannot, however, be sufficient. Maternal health is influenced by the complex interaction of multiple social factors. Among these factors, socio-economic determinants, such as migrant women's administrative status, should be considered and analyzed as potential intermediate factors. Race/ethnicity, a major determinant in contexts such as the USA or Australia, even when migration is the main focus of the studies, should also be collected and included in the analysis to enable multidimensional descriptions of the groups being compared and provide deeper insight to the causal relationships with health outcomes.

To our knowledge, this is the first systematic review and meta-analysis focusing on the association of migrant women and native-born women in high-income countries with various severe maternal outcomes and according to a single and consensual definition of migrant women. In contrast to previous systematic reviews, we have included all data available from high-income countries and not only those in Europe [8,35]. This offers the opportunity to compare national contexts with various migration demographic patterns, as well as various

public policies related to migration. Use of this single, consensual definition makes it possible to address migration issues specifically, separate from overall ethnic or racial disparities, although the ethnic/racial dimension is probably one of the mechanism involved in the disparities faced by migrant women [36,45,46]. It also reduced the heterogeneity between studies and improved their comparability. This important heterogeneity in the definitions of migrant women across studies was the main limitation of a previous review and meta-analysis [8]. This definition of migrant women, as being born outside the host country, is the one used by the International Organization for Migration [84] and recommended by the Reproductive Outcomes and Migration collaboration and the EURO-PERISTAT project, to study migrants' perinatal health [36]. We were able to apply this definition for all included studies except for some Dutch studies [9,65,66] where we cannot exclude that some second-generation migrant women were included in the migrant women group. This definition offers the advantage of targeting a homogeneous group of first-generation migrants. Second-generation migrant women constitute a specific group interesting to study, but databases with available data on the parents' and grandparents' countries of birth are still too rare for a synthesis of their results to be useful. Our planned subgroup analyses enabled us to explain a part of the still substantial heterogeneity between studies. In order to explore between-study heterogeneity, we decided to perform subgroup analyses instead of meta-regression with host country and region of birth as moderators because these characteristics are closely related in the set of studies we found, resulting in collinearity in the model, likely to bias the estimates. As a consequence, we cannot totally rule out that host country and region of birth actually characterized the same subgroups of migrant women. Nevertheless, our analysis was still limited by the persistent heterogeneity in definitions of all-cause and cause-specific severe maternal morbidity. Moreover, our review may be limited by the questionable validity of administrative databases because of the uncertain quality of the reporting and the intrinsic limitations of ICD codes—both sources of potential misclassification [43]. Another limitation is that we could not take into account other social factors that may interact with migration to determine maternal health, because many of these data are unavailable. The paucity of studies with data on migrant women's legal status prevented a subgroup analysis of this variable. Finally, we cannot exclude the risk of selective outcome reporting bias within studies. Researchers may report their findings selectively by choosing to focus on selected outcomes and analyses based on the results [51].

Our findings provide valuable insights for further exploration of the causal mechanisms of maternal health inequalities by highlighting those faced by migrant women in high-income countries according to both their region of origin and the host country. Our analysis was still limited by the persistent heterogeneity in definitions of all-cause and cause-specific severe maternal morbidity. Thus, future studies should use harmonized definitions for all-cause and cause-specific severe maternal morbidity, and future investigations should examine the extent to which specific hypothesized mechanisms, related to the characteristics of the women or the healthcare system, explain the association between migrants and severe maternal outcomes in different settings. Approaches combining the joint characterization and analysis of severe maternal outcomes according to ethnicity, race, and migration could shed light on these question and help to guide policies and care.

To conclude, the differential risk of severe maternal outcomes between migrant women and native-born women in high-income countries varies by the host country and the migrant woman's region of birth. Our findings highlight the maternal health inequalities faced by migrant women in high-income countries according to their region of origin and host country. Future studies should use harmonized definitions for all-cause and cause-specific severe maternal morbidity and be designed to explore in greater depth the mechanisms of inequalities between migrant women and native-born women.

## Supporting information

**S1 Table. Literature search algorithm on Medline via PubMed.**
(DOCX)

**S2 Table. Literature search algorithm on Embase.**
(DOCX)

**S3 Table. Literature search algorithm on Cochrane Library.**
(DOCX)

**S4 Table. List of the excluded studies and reasons for their exclusion.**
(DOCX)

**S5 Table. Characteristics of the included papers, listed in alphabetical order of first authors.**
(DOCX)

**S6 Table. Methodological quality of the studies included (listed in alphabetical order) assessed with the Newcastle–Ottawa Scale (NOS).**
(DOCX)

**S7 Table. Amendments made to the protocol PROSPERO.**
(DOCX)

**S1 Fig. Maternal mortality in migrant women and native-born women.**
(DOCX)

**S2 Fig. All-cause severe maternal morbidity in migrant women and native-born women.**
(DOCX)

**S3 Fig. Maternal intensive care unit admission in migrant women and native-born women, stratified by host country.**
(DOCX)

**S4 Fig. Maternal intensive care unit admission in migrant women and native-born women stratified by migrant women's region of birth.**
(DOCX)

**S5 Fig. Funnel plot for studies reporting maternal mortality overall.**
(DOCX)

**S6 Fig. Funnel plot for studies reporting maternal mortality stratified by host country.**
(DOCX)

## Acknowledgments

The authors would like to thank Annika Esscher and Birgitta Essén (both from the Department of Women's and Children's Health, Uppsala University, Uppsala, Sweden) who supplied data about the number of deaths and maternal mortality ratios for women of reproductive age born in low-, middle-, and high-income countries and in Sweden; Ayesha Siddiqui (Université de Paris, CRESS, Obstetrical Perinatal and Pediatric Epidemiology Research Team, EPOPé, INSERM, INRA, Paris, France) who supplied data about the total number of women diagnosed with severe preeclampsia by distinguishing the women born in France from those born elsewhere in Europe; Baiju Shah (Sunnybrook Health Sciences Centre, Toronto, Ontario, Canada; Sunnybrook Research Institute, Toronto, Ontario, Canada; Department of Medicine,

University of Toronto, Ontario, Canada) who supplied data for the total number of women with and without preeclampsia/eclampsia among each of the nonimmigrant, refugee, and other immigrant groups; Birgit Reime (Faculty of Health Safety Society, Furtwangen University, Furtwangen, Germany) who supplied data for the total number of women diagnosed with cause-specific severe maternal morbidity among native-born and migrant women in Germany between 2001 and 2007; Jessica Turner (Mater Research Institute, University of Queensland, Brisbane, Australia) who supplied data for the total number of women diagnosed with massive postpartum hemorrhage and ICU admission among refugees and women born in Australia; Joost Zwart (Department of Obstetrics and Gynaecology, Deventer Ziekenhuis, 7416 SE Deventer, the Netherlands) and Jos van Roosmalen (Department of Obstetrics and Gynaecology, Leiden University Medical Center, 2300 RC Leiden, the Netherlands) who supplied data for the total number of deliveries among Western and non-Western migrant women; Kjersti Sletten Bakken (Centre for Intervention Science in Maternal and Child Health [CISMAC], Department of Global Public Health and Primary Care, University of Bergen) who supplied data about the total number of women diagnosed with preeclampsia/eclampsia and HELLP syndrome among first-generation immigrants, second-generation immigrants, and native Norwegian-born women; Marcelo Urquia (Department of Community Health Sciences, University of Manitoba Faculty of Health Sciences, Winnipeg, Manitoba, Canada) who supplied data about the total number of women diagnosed with eclampsia according to their region of birth; Miguel-Angel Luque-Fernandez (Department of Non-Communicable Disease Epidemiology, Cancer Survival Group, London School of Hygiene and Tropical Medicine, London; and Department of Non-Communicable Disease and Cancer Epidemiology, Instituto de Investigacion Biosanitaria de Granada (Ibs.GRANADA), University of Granada, Granada, Spain) who supplied data about the total number of live births in Spain and in Europe excluding Spain; Stephanie Leonard (Department of Obstetrics and Gynecology, Stanford University, Palo Alto, California; California Maternal Quality Care Collaborative, Stanford University, Palo Alto, California) who supplied data on the total number of women with severe maternal morbidity, and with non-transfusion severe maternal morbidity in native-born women and migrant women. We would also like to thank Jo Ann Cahn for her English editorial assistance.

## Author Contributions

**Conceptualization:** Maxime Eslier, Elie Azria, Konstantinos Chatzistergiou, Zelda Stewart, Agnès Dechartres, Catherine Deneux-Tharaux.

**Data curation:** Maxime Eslier, Konstantinos Chatzistergiou, Zelda Stewart.

**Formal analysis:** Maxime Eslier, Elie Azria, Konstantinos Chatzistergiou, Agnès Dechartres, Catherine Deneux-Tharaux.

**Investigation:** Maxime Eslier, Elie Azria, Catherine Deneux-Tharaux.

**Methodology:** Maxime Eslier, Elie Azria, Konstantinos Chatzistergiou, Zelda Stewart, Agnès Dechartres, Catherine Deneux-Tharaux.

**Project administration:** Maxime Eslier.

**Software:** Maxime Eslier, Konstantinos Chatzistergiou, Agnès Dechartres.

**Supervision:** Maxime Eslier, Elie Azria, Agnès Dechartres, Catherine Deneux-Tharaux.

**Validation:** Maxime Eslier, Elie Azria, Konstantinos Chatzistergiou, Agnès Dechartres, Catherine Deneux-Tharaux.

**Visualization:** Maxime Eslier, Elie Azria, Konstantinos Chatzistergiou, Zelda Stewart, Agnès Dechartres, Catherine Deneux-Tharaux.

**Writing – original draft:** Maxime Eslier, Elie Azria, Catherine Deneux-Tharaux.

**Writing – review & editing:** Maxime Eslier, Elie Azria, Konstantinos Chatzistergiou, Zelda Stewart, Agnès Dechartres, Catherine Deneux-Tharaux.

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
