## [Editor Report · Decision Letter 0]

21 Sep 2022

Dear Dr ESLIER, 

Thank you for submitting your manuscript entitled "Association between migration and severe maternal outcomes in high income countries: systematic review and meta-analysis" for consideration by PLOS Medicine.

Your manuscript has now been evaluated by the PLOS Medicine editorial staff and I am writing to let you know that we would like to send your submission out for external peer review.

Please re-submit your manuscript within two working days, i.e. by Sep 23 2022 11:59PM.

Kind regards,

Beryne Odeny

PLOS Medicine

---

## [Decision Letter · Decision Letter 1]

17 Apr 2023

Dear Dr. ESLIER,

Thank you very much for submitting your manuscript "Association between migration and severe maternal outcomes in high income countries: systematic review and meta-analysis" (PMEDICINE-D-22-03069R1) for consideration at PLOS Medicine. 

[LINK]

In light of these reviews, I am afraid that we will not be able to accept the manuscript for publication in the journal in its current form, but we would like to consider a revised version that addresses the reviewers' and editors' comments. Obviously we cannot make any decision about publication until we have seen the revised manuscript and your response, and we plan to seek re-review by one or more of the reviewers. 

We expect to receive your revised manuscript by May 08 2023 11:59PM. Please email us (plosmedicine@plos.org) if you have any questions or concerns.

We look forward to receiving your revised manuscript. 

Sincerely,

Philippa Dodd MBBS MRCP PhD

PLOS Medicine

plosmedicine.org

GENERAL

Please respond to all editor and reviewer comments detailed below, in full.

* Please revise your use of language when referring to both ‘migrants’ and ‘natives’, suggest instead native-born women and migrant women

** Please check carefully for the use of appropriate grammar, a number of minor errors negatively impacting reader accessibility were identified, including by the reviewers. We suggest inviting a native English speaker to proof read the manuscript prior to re-submission. Should the manuscript proceed successfully through the peer review process to publication, our copyeditors can help you further.

*** Please update your search to the present time. PLOS Medicine requires that all SR/MAs are updated to within 6 months of an anticipated publication date

COMMENTS FROM THE ACADEMIC EDITOR

I agree that this is an important area. However, as presented, the findings do not add a huge amount to the literature. In the main this is due to the simplistic comparison of migrant and native, as pointed out by Reviewer 4. .

It seems appropriate to challenge the authors to provide a major revision, including reanalyses as suggested by the statistical reviewer, and more nuanced discussion of the limitations as suggested by the other reviewers.

ABSTRACT

Thank you for reporting your abstract according to PRISMA for abstracts, following the PLOS Medicine abstract structure (Background, Methods and Findings, Conclusions) 

We note reviewer #1 comments (below) and agree with the authors regarding combining the methods and findings section i-line with our formatting requirements.

Line 41 – please define RR and 95% CI at first use

Suggest the use of commas instead of hyphens (as these can be confused with reporting of negative values) to separate upper and lower bounds 

PLOS Medicine requests that where 95% CIs are reported p values are also reported. Please report a p <0.001 or where higher as p=0.002, for example. Suggest formatting statistical information as follows, “…(pooled RR 1.36; 95%CI [1.13-1.64]; p</=)…”

Please include any important dependent variables that are adjusted for in the analyses

In the last sentence of the Abstract Methods and Findings section, please describe the main limitation(s) of the study's methodology.

Abstract Conclusions:

Please address the study implications without overreaching what can be concluded from the data; the phrase "In this study, we observed ..." may be useful.

Please interpret the study based on the results presented in the abstract, emphasizing what is new without overstating your conclusions.

Please avoid vague statements such as "these results have major implications for policy/clinical care". Mention only specific implications substantiated by the results.

When making your revision please ensure avoidance of any assertions of primacy

AUTHOR SUMMARY

At this stage, we ask that you include a short, non-technical Author Summary of your research to make findings accessible to a wide audience that includes both scientists and non-scientists. The author summary should consist of 2-3 succinct bullet points under each of the following headings:

• Why Was This Study Done? Authors should reflect on what was known about the topic before the research was published and why the research was needed.

• What Did the Researchers Do and Find? Authors should briefly describe the study design that was used and the study’s major findings. Do include the headline numbers from the study, such as the sample size and key findings. 

• What Do These Findings Mean? Authors should reflect on the new knowledge generated by the research and the implications for practice, research, policy, or public health. Authors should also consider how the interpretation of the study’s findings may be affected by the study limitations.

The Author Summary should immediately follow the Abstract in your revised manuscript. This text is subject to editorial change and should be distinct from the scientific abstract. Please see our author guidelines for more information: https://journals.plos.org/plosmedicine/s/revising-your-manuscript#loc-author-summary

METHODS and RESULTS

Please include details of how non-English language sources of studies were handled – see also methodological reviewer comments attached.

Could your study have searched databases more widely, would this have provided nay helpful additional information?

Line 167 – “…databases…” we agree with the methodological reviewer that seems rather an inappropriate description, suggest “…studies…”, as an alternative 

As above, PLOS Medicine requests that where 95% CIs are reported p values are also reported. Please report a p <0.001 or where higher as p=0.002, for example. Please format statistical information as suggested above for the abstract.

Please replace the use of hyphens with commas to separate upper and lower bounds of 95% CIs

TABLES

Table 1 – please clarify/revise the use of the term vital statistics

FIGURES

Figure 1 – title: suggest “…study selection…”

As above, please ensure to revise figure titles to read “native-born”

Throughout, please indicate in the figure captions whether your analyses are adjusted or unadjusted. Where adjusted analyses are presented, please also include unadjusted analyses for comparison and clearly state the factors that are adjusted for. 

Please also clearly indicate the meaning of the dots and lines in the figures. All captions should clearly report the figure content without the reader needing to refer to the manuscript text. 

DISCUSSION

Please remove all sub-headings from the discussion such that the discussion reads as continuous prose.

Please present and organize the Discussion as follows: a short, clear summary of the article's findings; what the study adds to existing research and where and why the results may differ from previous research; strengths and limitations of the study; implications and next steps for research, clinical practice, and/or public policy; one-paragraph conclusion. 

Line 230 – please avoid assertions of primacy which can be risky, suggest “…to our knowledge…”

REFERENCES

Please include an access date for web references

SUPPORTING INFORMATION

Please ensure that figure/table captions clearly define all content for the reader without the need to refer to the text

PRISMA checklist – thank you for including the PRISMA checklist, please amend and refer to section and paragraph numbers rather than page (or line) numbers as these often in the event of publication.

Table S3 – in context of transparent data reporting, the editorial team hugely appreciated the inclusion of this table but it is not a requirement.

Table S5 – please clarify/revise the use of the term vital statistics

Comments from the reviewers:

Reviewer #1: The authors address an important public health issue. It is nicely written, apart from the numerous unnecessary Oxford-commas. I would be interested to hear from the authors whether ´inequities` are not better than `inequalities`! I would separate `methods` from `findings´ in the abstract. 

My main problem is with the definition of migrants as those who were not born in the host country. What then do we do with `women`who are born in the host countries`but with parents from the country of origin? Are they not migrants anymore?

That is different what we did with similar studies in the Netherlands. The definition also confused me in line 45, 282, 296 and 312 and that should be addressed in a revised version. 

I agree with you that you cannot consider all migrants (called global migrants) as coming from the same background and consider them as coming from different regions is a helpful concept. 

In abstract, line 50/51 it is unclear when you state `there is a need .... and to customize related health policies in each context.` Customized to what?

In line 51/52 you excluded maternal mental health problems from the review, but did not give an argument for that. It is, however, a frequent outcome of childbirth, and needs a reason for exclusion.

I did find in the figures one of our dutch papers being referenced as `Zwart and Van Hanegem`±I saw Zwart´s BJOG paper in the reference list but not the paper with Van Hanegem as the first autor (Acta Obstet Gynecol Scand).

Reviewer #2: See attachment

Michael Dewey

Reviewer #3: Overall, the study is well conducted and reported in a clear manner. I have the following comments for authors to address

1) Why did authors not used a method o describe the overall certainty of evidence such as GRADE methodology. Authors seems to focus on results of meta-analysis which is one piece of information and it needs to be considered with aspects of evidence such as type of study, risk of bias, heterogeneity of the evidence, indirectness and precision of summary estimates. 

2) Why did authors not report an overall risk of maternal mortality in high income country irrespective of region in high income country?

3) Authors report that risk of maternal mortality is high in immigrant mothers in Europe based on RR 1.36 (1.13-1.64). This mean a relative increase risk of about 36 %. What is the absolute risk per 100, 000 or number need to harm? This is important as the absolute risk might be small as overall the maternal mortality is not a common event. 

Reviewer #4: Dear authors, 

Thank you for the opportunity to review this manuscript. 

The manuscript offers important contributions to the field, as the research question is relevant to discussions regarding perinatal health disparities between migrants from different origins and non-migrants in different host countries. The meta-analyses are performed in line with current epidemiological standards and methods and findings are presented clearly.

However, I do believe that the research question and analysis are limited by the unidimensional comparison of foreign-born vs native-born women globally. Maternal mortality/morbidity is determined by complex interactions of a wide range of (social) determinants which need to be integrated in (intersectional) analyses to advance the current state of knowledge on this topic. Especially with regards to the Americas, comparing "native" to "foreign"born populations does not do justice to the colonial history, migration background of now 'native' populations etc. In line with this critique on the comparisons made, I believe that exclusion of studies before 1990 only has a very small impact on limiting heterogeneity in migrants' characteristics and integration policies.

Therefore, at least in the discussion of the findings, more attention needs to be given to the fact that comparing the umbrella term of migrants to the umbrella term of natives - without considering race/ethnicity of native groups as well, along with other determinants and their interactions - has only limited value to draw conclusions regarding the causal relationships between migration, ethnicity or any other variable and outcomes. For example, the authors could rephrase claims such as "the risk of severe maternal complications for mothers in the USA/Australia appears to be related more to race/ethnicity than to migration status". The authors should also be conscious of consistent use of all terms throughout the manuscript (for example, earlier in the paper they refer to "migration status" meaning "legal status", but in this specific statement migration status seems to refer to the fact that someone is a first-generation migrant).

With expanded critical reflections in the discussion of the findings, I believe the article to be of interest to clinicians and policymakers in the field of perinatal healthcare, as well as to researchers in the field of migration-related health inequities. 

Other points of concern for minor revisions:

- In the presentation of the finding concerning differential risk of severe outcomes in women from different regions of origin, it is now unclear that this analysis was performed for studies both in USA and Europe. From the reading of the results, the reader could assume that this difference only concerns studies from Europe (given that no differences were found between migrants-natives in USA without taking country of birth into account). 

- The manuscript is generally well-organized and written clearly enough to be accessible, although it would benefit from a thorough language check by a native English speaker given a number of (relatively minor) mistakes that cumulatively do compromise easy reading of the review.

[LINK]

---

## [Decision Letter · Decision Letter 2]

26 May 2023

Dear Dr. ESLIER,

Thank you very much for re-submitting your manuscript "Association between migration and severe maternal outcomes in high income countries: systematic review and meta-analysis" (PMEDICINE-D-22-03069R2) for review by PLOS Medicine.

I have discussed the paper with my colleagues and the academic editor and it was also seen again by 3 reviewers. I am pleased to say that provided the remaining editorial and production issues are dealt with we are planning to accept the paper for publication in the journal.

[LINK]

We look forward to receiving the revised manuscript by Jun 02 2023 11:59PM.   

Sincerely,

Philippa Dodd, MBBS MRCP PhD

PLOS Medicine

plosmedicine.org

Requests from Editors:

GENERAL

Thank you for your detailed and considered responses to previous editor and reviewer requests. Please see below for further comments which we require you address prior to publication.

Comments from the Editor-in-chief: The finding of no difference in outcomes in the US (or Australia) is interesting and surprising. How different is the benchmark across these countries i.e. are maternal outcomes for native-born women worse in the US (and/or Australia) compared to outcomes for native-born women in Europe, for example? Please consider how the results are framed/compared in this context.

ABSTRACT and AUTHOR SUMMARY

It would be helpful to indicate that the studies included were conducted at a national level.

AUTHOR SUMMARY

Line 84 – ‘This result provides insight valuable for further exploring these inequalities’ mechanisms.’ Suggest instead, ‘These data highlight the need to further explore the mechanisms underlying these inequities.’ Or similar

Line 86 – the latter part of this statement is rather vague ‘…take into account the multiple dimensions of the women’s social context’ please revise. 

TABLES

Table S3 – please feel free to include this table with the manuscript as supporting information.

REFERENCES

Ref 38 line 523 – what does the asterisk represent here? Can it be removed?

Ref 67 line 608 – what to the 3 stars represent here? Can they be removed?

SOCIAL MEDIA

If not already done so, to help us extend the reach of your research, please detail any Twitter handles you wish to be included when we tweet this paper (including your own, your co-authors’, your institution, funder, or lab) in the manuscript submission form when you re-submit the manuscript.

Comments from Reviewers:

Reviewer #1: The authors have undertaken a huge job to review all the literature in relation to migration and maternal health and bring together these data in a nice review. There is only one issue I still have, because my earlier question was not answered. I have very well understood their defintion of migrant and native women. The Dutch data, however, are still a bit different, because in our papers a woman born in the Netherlands but with one or two parents born in Marocco is still considered as a migrant woman. Only in the paper about asylumseekers the migrants cannot be born in the Netherlands, so they all confirm with the definition used in the paper. I cannot assess this issue for the other studies. But I would suggest to include this as a limitation, at least for our dutch studies. Jos van Roosmalen 

Reviewer #2: The authors have addressed all my points but there is one minor issue remaining.

I suggested performing a meta-regression using both host country and birth region as moderators. I did not find the response very convincing. They will no doubt be related, that is the point, but I do not see that the analysis is necessarily fatally flawed a priori. I would suggest trying it and seeing what happens or at least expand the limitations section to indicate that the relationship with host country and the relationship with birth region may be saying the same thing twice.

Michael Dewey

Reviewer #4: Dear authors, 

It was a pleasure to review your revised manuscript. I believe you improved the manuscript substantially based on all reviewers' comments. The results are more meaningful and the discussion offers more depth and critical reflection now. I think the manuscript is now acceptable for publication, although I still think the quality and clarity of the writing could be improved in some instances, e.g. line 273 "Our systematic review shows that (...), while women in the USA and Australia did not." should be "while MIGRANT women in the USA and Australia did not". 

Best regards

[LINK]

---

## [Editor Report · Decision Letter 3]

5 Jun 2023

Dear Dr ESLIER, 

On behalf of my colleagues and the Academic Editor, Dr. Sarah Stock, I am pleased to inform you that we have agreed to publish your manuscript "Association between migration and severe maternal outcomes in high income countries: systematic review and meta-analysis" (PMEDICINE-D-22-03069R3) in PLOS Medicine.

PRESS

Best wishes,

Pippa 

Philippa Dodd, MBBS MRCP PhD 

PLOS Medicine